# From Robot Learning To Robot Understanding: Leveraging Causal Graphical Models For Robotics

**Kaylene Stocking, Alison Gopnik, Claire Tomlin**
University of California, Berkeley
{kaylene, gopnik, tomlin}@berkeley.edu

**Abstract:** Causal graphical models have been proposed as a way to efficiently and explicitly reason about novel situations and the likely outcomes of decisions. A key challenge facing widespread implementation of these models in robots is using prior knowledge to hypothesize good candidate causal structures when the relevant environmental features are not known in advance. The tight link between causal reasoning and the ability to intervene in the world suggests that robotics has much to contribute to this challenge and would reap significant benefits from progress.

**Keywords:** causal reasoning, cognitive science, causal graphical models

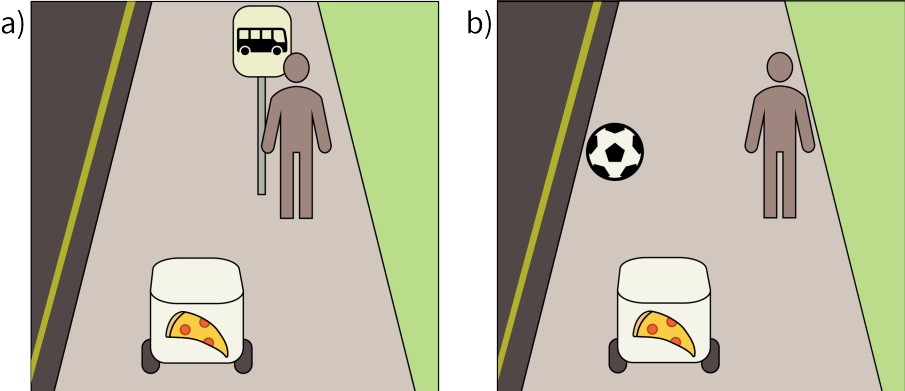

Figure 1: A pizza delivery robot faces two scenarios that machine learning methods may struggle to differentiate and handle appropriately. In panel (a), continuing to drive is is a reasonable action; in panel (b), slowing down or even stopping is necessary for safety. Arriving at the correct decision requires integrating combinations of observations that are unlikely to have occurred jointly in training data. Causal hypotheses, inspired by cognitive science research, may help autonomous systems make good, interpretable decisions in novel situations.

## 1 Introduction

The difficulties of bringing autonomous driving technology to fruition have highlighted a longstanding central challenge for robots that aim to operate in uncontrolled environments: how to handle novel situations not represented in training data. While gathering more data may help with some of these situations, there are diminishing returns as we consider rarer edge cases. For example, consider the situation illustrated in Figure 1, where a pizza delivery robot wants to avoid bumping into a human standing on the sidewalk. The human is currently standing still, but if they suddenly decide to move, it may not be possible for the robot to stop in time. Overly conservative behavior such as never passing pedestrians may result not only in cold pizza but reduced robot predictability for other people on the sidewalk. Therefore, it is highly desirable for the robot to be able to make an accurate prediction of the human's intended behavior.

Blue Sky Papers, 5th Conference on Robot Learning (CoRL 2021), London, UK.

Humans can look at the two panels of Figure 1 and immediately generate relevant and likely hypotheses about the pedestrian's behavior. In the first panel, the presence of a bus stop strongly suggests that the person is waiting for the bus, and with no bus in sight, is unlikely to move. In the second, a soccer ball in the sidewalk may have been kicked or dropped by the human. If they intend to retrieve it, slowing down is clearly the best choice. Replacing the soccer ball with a basketball should result in the same hypothesis and decision, but replacing it with a piece of trash should not. Humans can effortlessly reason about and generalize across these simple variations, but statistical machine learning models may struggle to handle novel combinations.

Causal graphical models, inspired by how humans learn, have been proposed as an efficient way to probabilistically reason about uncertain situations [1]. In section 2 we discuss the opportunity and motivation for explicit causal reasoning in robotics. Then, in section 3 we return to the motivating example and describe how current work on causal inference allows for useful inference in robotics problems, as well as early strategies that have been proposed for the crucial and difficult step of hypothesizing candidate causal structures.

## 2    Causal Reasoning and Robotics

Causal graphical models (CGMs) explicitly encode the effects of an agent's interventions, making them especially appropriate for reasoning about possible actions and making decisions. Furthermore, they allow the agent to give explanations for its behavior and estimate its uncertainty about outcomes, both of which are important for safe interactions with humans but difficult to achieve with "black-box" models such as deep neural networks. To effectively leverage CGMs in novel scenarios, agents must be able to generate relevant causal hypotheses and link them to past and current observations. The ability to do this even in limited domains such as making predictions about human trajectories would be invaluable for robots that need to operate in complex naturalistic environments or alongside people.

If the field of robotics has much to gain from the causal reasoning paradigm, it also has much to contribute. From a practical perspective, formalizing causal models of learning and decision-making to create practically useful algorithms will require simulated environments and tasks that simplify or abstract away some challenges but are complex enough to require abstract causal reasoning. Simulated benchmarks like operating MuJoCo robots [2] and playing Atari games [3], in which agents do not know which features are important for success *a priori*, have helped fuel an explosion in improvements in reinforcement learning. Meanwhile, causal learning has demonstrated effectiveness at inherently causal tasks such as interpreting medical treatment outcomes, but the relevant variables and constraints on graph structures are typically provided by human designers in advance [4]. Roboticists can help develop benchmarks where discovering the correct features for building a useful causal model for some domain - in other words, the hypothesis generation problem - is non-trivial. Furthermore, to interact productively with these more complex environments, successful agents will most likely need to incorporate techniques from robotics including control theory, reinforcement learning, and/or state estimation.

From a more philosophical perspective, there is an intuitively appealing link between causal reasoning and robotics as the study of embodied intelligence. Experiments in animals suggest that the ability to intervene on the world and observe the consequences is fundamental to biological learning [5]. Similarly, there is a large body of evidence supporting the prominence of causal predictions in the human neural and cognitive architecture [6, 7]. Causal knowledge is critically important to biological intelligence, and yet it can be proven that the associational relationships learned by classical statistical and machine learning algorithms are in general insufficient to build accurate causal models [8]. To understand the world the way humans do, agents will need the ability to interact with it like we do - and this is exactly why research in causal cognition should be tightly linked with efforts to design robots that can learn and utilize causal information about their environments.

A growing number of disciplines have begun leveraging causal graphical models [9, 4, 10]. In robotics and automation, models that are pre-specified (e.g. by human experts) have already found use in applications such as imitation learning [11], calculating the probabilities of complex failure modes [12], generating task hierarchies that allow for easier policy optimization [13], and generalizing learned policies to modified tasks [14]. A limited form of learned CGM that only considers observable direct causes of an expert's actions has been successfully applied in imitation learning

[15]. Algorithm design for learning more general CGMs from observations [16, 17], interventions (actions), or combinations thereof [18] is an active area of research. However, these algorithms generally require many samples from a static environment. Robots that hope to leverage causal models in novel situations, meanwhile, will need to utilize prior knowledge to generate new causal hypotheses given a limited set of observations. For large state spaces they will also need to identify which environmental states or features to include in causal models. Formalizing this may be challenging, but the growing body of research on apparent human use of CGMs supports their practical utility [19, 20, 21], and recent promising ideas on how they might be generated suggest several exciting avenues for future research.

## 3 Generating and Reasoning with Causal Graphical Models

CGMs consist of a set of variables linked by directed causal relationships, where intervening to change the value of one variable probabilistically changes the value of all nodes "downstream" of it (i.e., its effects). We refer the reader to prior work for a more detailed overview [1, 20]. Using CGMs in problems such as the pizza delivery motivating example can be broken down as follows:

1. Generate possible causal models with nodes representing variables relevant to the current situation, linked by causal relationships learned from past experience.

2. Use the values of observed variables to evaluate the probability of each model being a good fit for the current situation.

3. Select the most likely model and use it to efficiently infer future states and the possible consequences of decisions, represented as interventions on appropriate variables in the model.

The last two steps have been formalized as expressive and efficient probabilistic algorithms [1], but the first step, causal hypothesis generation, is a young and active area of research. In the remainder of this section, we describe how each step might be approached.

### 3.1 Hypothesis generation as search

Where do human causal hypotheses come from? Several intriguing leads have been proposed by cognitive scientists studying learning. A common thread between many of these ideas is to frame hypothesis generation as a stochastic search problem. If an agent can represent (perhaps implicitly) a probability distribution over all the possible hypotheses for a given problem, then sampling from this distribution would result in a valid set of different hypotheses to consider. Whether this approach would be successful in practice hinges critically on how good (in the sense of accuracy and decision-making utility) the best hypothesis in the candidate set is. To obtain better candidates with fewer samples, we could either find ways to constrain the search space by eliminating impossible or very unlikely hypotheses, or shape the prior distribution of hypotheses to sample with greater frequency those that are more likely to be valid. Both of these ideas have been explored in the literature.

### 3.1.1 Tuning the hypothesis prior distribution

One approach to shaping the hypothesis distribution is inspired by the observation that it is perfectly possible to have a hypothesis about what which types of hypotheses are likely in a particular scenario. This higher-level hypothesis about what *kinds* of ideas are relevant is commonly called an "overhypothesis" [22]. This begs the question - if the "base level" hypotheses can be evaluated and selected using Bayesian inference procedures, why not overhypotheses as well? This idea is especially appealing in that it requires no new machinery over what is already necessary for working with the base-level hypotheses. It can be formalized as a hierarchical Bayesian model (HBM), in which learning multiple "tiers" of hypotheses may not even require any more environmental observations than learning only a base-level hypothesis [23, 24].

Since real-world systems must operate with finite memory and computational resources, it is unlikely that prior distributions of hypotheses could be maintained for every kind of scenario. Surprisingly, experimental data supports the idea that humans may sample from a single distribution when performing a large class of cognitive tasks called modal reasoning, which encompasses judgments about what is possible, probable, moral, and so on [25]. Human judgments in different tasks are more highly correlated under time pressure, suggesting that the first possibilities that come to mind

(the most likely ideas to be sampled) are independent of the specific task [26]. Therefore, storing even a small number of prior distributions may still have practical utility.

An alternative to prior distributions stored in memory would be to generate a new one *ad hoc* for each new task. For example, features of the task could bring to mind solutions encountered in the past and "warm start" the search process [27]. It might also be desirable to take a candidate hypothesis and modify it, rather than sampling an entirely new one. Here, too, insights from cognitive science may be helpful. The "child as hacker" paradigm highlights the goal-oriented nature of human cognition, and notes that when humans "hack" on a problem, we often consciously aim for a solution that is better along one of many possible dimensions, such as accuracy, simplicity, or even novelty [28]. Focusing on improvements in a particular dimension could constrain the types of solution that are considered; for example, aiming for simplicity might eliminate theories that involve any additional variables. Switching between goals effectively splits the daunting task of coming up with the best theory into smaller chunks that lead to gradual improvements in the overall quality of the working hypothesis.

### 3.2 Causal inference

In the event that likely and relevant hypotheses can be generated successfully, utilizing them is straightforward. Two possible hypotheses for the pizza delivery problem are illustrated in Figure 2, where we use binary-valued variables for simplicity. H represents a hypothesis about what the human is doing, R represents whether the human is about to run across the sidewalk, and S represents whether the outcome is safe (the robot doesn't bump into anyone). The robot's ability to choose an action is represented by A. Panel (a) shows a hypothesis that the human is waiting for a bus, with a causal link to being near a bus stop and a low probability of the human running across the sidewalk. Panel (b) represents an alternative hypothesis that they're playing with a ball, yielding a high probability of running.

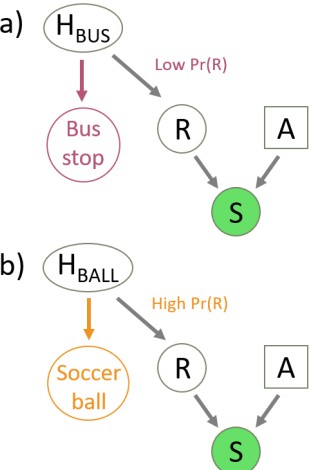

Figure 2: Two hypotheses for the pizza delivery example.

To perform step 2, if the robot observes a bus stop and not a soccer ball, it can use the graphs to infer that the first hypothesis is more likely and the second one less likely than their prior probabilities (e.g. the fraction of time pedestrians engage in each behavior). Equivalent calculations can be performed in the alternative scenario where the robot observes a ball instead of a bus stop. Finally, a forward propagation calculation for each hypothesis easily yields estimates of the probability of safety given that the robot decides to stop or keep going (step 3). Subsequent observations of the human's behavior can be used to both refine the relevant priors as well as shape the distribution of which hypotheses should be considered in similar future situations.

The inferences described above are efficient and allow for an explicit, interpretable calculation of the best action. The example model presented here does rely on significant amounts of prior knowledge, such as the fact that people usually stand by bus stops because they are waiting for a bus. However, only estimates about the direct relationship between each pair of nodes need to be provided. The graph structure takes care of composing individual causal relationships that are commonly encountered into a reasonable probability estimate for the outcome of a situation that may be completely new to the robot.

## 4 Conclusion

Algorithms for generating plausible causal hypotheses in naturalistic environments would enable leveraging the powerful machinery of causal inference for many practical robotics problems, including making interpretable decisions about novel situations. The close link between embodied intelligence and causal reasoning suggests an important opportunity for roboticists to contribute to progress in this area, including by designing appropriate benchmarks and integrating causal learning with algorithms for perception and successful interventions in the world.

**Acknowledgments**

We would like to thank the anonymous reviewers for their helpful comments.

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
