# OpenReview forum: "From Robot Learning To Robot Understanding: Leveraging Causal Graphical Models For Robotics"
_robot-learning.org/CoRL/2021/Conference/Blue_Sky — CoRL 2021, Blue Sky_

### Official Review · Reviewer_kQHh · 2021-07-31

**Novelty:** Very Good
**Impact:** 4
**Clarity Of Presentation:** Very Good

**Recommendation:**

Strong Accept: I recommend accepting the paper and will argue for my recommendation even if other reviewers hold a different opinion.

**Summary:**

This paper highlights the promise of leveraging causal models in robotics. The paper argues that reasoning with causal models could allow robots to generalize more effectively to novel scenarios. More specifically, the paper argues that a key challenge in this area is to develop approaches for generating candidate causal hypotheses when a robot is faced with a novel environment. Cognitive science-inspired approaches to this problem are outlined, along with concrete directions for future efforts.

**Summary Of Recommendation:**

Overall, this paper provides a novel and interesting perspective on the use of causal models in robotics. While there is a growing body of work on leveraging ideas from causality in supervised learning, there is relatively little work in this area within robotics. At a high-level, the paper clearly and convincingly articulates the argument that robotics applications motivate specific technical challenges in this area. More specifically, the paper highlights the concrete problem of automatically generating causal hypotheses when a robot is faced with a novel environment. This is an important problem that is likely to generate follow-up work. The paper also provides concrete ideas inspired by the literature in cognitive science. These ideas are promising and could spur additional research in this area. Overall, the paper highlights an interesting and under-studied problem and also proposes concrete ideas to make progress. As a result, I believe that the paper has the potential to make an important contribution in the long-term.

---

### Official Review · Reviewer_kSXB · 2021-08-25

**Novelty:** Good
**Impact:** 3
**Clarity Of Presentation:** Very Good

**Recommendation:**

Weak Reject: I recommend rejecting the paper, but will not argue for my recommendation if the majority of other reviewers have a different opinion.

**Summary:**

This paper discussed the use of causal learning, as an alternative to data-centric approaches, to improve a robot's ability to reason about novel situations and make intelligent decisions. This manuscript pinpointed the synergetic relations between causal reasoning and robotics: on one side, causal learning tools will facilitate the robots to make informed decisions; on the other hand, the physical embodiment of robots will facilitate the discovery of appropriate causal models. The authors chose to use the classical Causal Graphical Models (CGMs) as the representation of the causal model, and discussed strategies to learn and make use of these CGM models for robotic problems.

**Summary Of Recommendation:**

This paper touched upon an important subject of causal learning for robotics. There has been substantial interest in the community in departing from pure data-driven approaches (e.g., end-to-end deep learning) to embracing model-driven paradigms for building more robust robotic systems. The authors' choice of causal reasoning machinery makes intuitive sense as a robotic problem can be naturally cast as a causal inference problem: The robot's actions purposefully move the current state of the world to a new state.

While there is a general consensus in the robot learning community that the ability to conduct causal reasoning is vital for improving a robot's systematic generalization and robustness in real-world situations, especially for out-of-distribution or rare events, it remained unsettled what the best forms of representations and computational tools are to operationalize causal reasoning in realistic robotics tasks.

This work motivated the conventional CGMs and causal inference algorithms with a toy example in Figure 1, where a few well-defined binary-valued variables are concerned. However, the applicability and scalability of such a method are unclear in real-world problems. This work cast hypothesis generation as a search problem, which could be intractable for large-scale problems with a great number of variables. This challenge of hypothesis generation is further exacerbated in the scenarios where:

1) The relevant variables are unknown and have to be identified (or discovered) from raw sensory signals from the robot;
2) The robot's actions generate unknown interventions, where their effects on the causal variables might not be explicitly defined.

Without the discussion on these practical challenges, this manuscript, unfortunately, did not pinpoint a visible pathway to operationalize causal learning in a tractable manner. While this is an exciting topic to work on, the reviewer would like to see more insights that the authors could bring to the table.

---

### Official Review · Reviewer_Lxpj · 2021-08-28

**Novelty:** Very Good
**Impact:** 4
**Clarity Of Presentation:** Very Good

**Recommendation:**

Strong Accept: I recommend accepting the paper and will argue for my recommendation even if other reviewers hold a different opinion.

**Summary:**

This paper argues that black-box, data-driven statistical machine learning approaches will, ultimately, be unable to display the kind of predictive reasoning that humans excel at, because they are fundamentally interpolators, and that causal reasoning is both necessary and well-suited to robotics. All of this is true. The authors also give a well-written and very accessible overview of the literature applying causal reasoning to robotics and sequential decision-making.

**Summary Of Recommendation:**

The argument that the paper makes about black-box machine learning is just inarguably true, and the the paper is well written and persuasive. The paper also points out that causal reasoning makes the most sense in areas where an agent is able to intervene to change the world; that's where it is easiest to test, in some sense, and also where it is most useful for making predictions.

One slight quibble I have with the whole line pushing causal reasoning into robotics and RL, is that it's never been clear to me how a causal model fundamentally differers from a transition model in RL; and we have been studying dependencies in forward models in a similar way since the early 2000s (primarily under the heading of factored MDPs). These approaches have also always used Bayesian modeling that explicitly considers similar relationships, e.g. they are often modeled using DBNs that are not fully connected. And of course there have been papers that try to learn which edges should be present and which not (e.g., Strehl in 2007 and Vigorito a few years later). It'd be helpful to clear up if these two lines are in some sense fundamentally the same, or if there's something that I (and many RL people) are missing.

I docked a point on clarity of presentation for the paper's use of numerical parenthetic citations as nouns, which spoiled an otherwise very pleasant reading experience.

---

### Decision · Program_Chairs · 2021-10-01

**Decision:**

Accept

**Comment:**

All reviewers agreed that the discussed topic of causal reasoning is very important for robotics and the paper is well written and persuasive. The authors should however add a discussion on how to use the proposed methods in setups where the relevant variables and effects are unknown and need to be inferred from sensory data.